# Deep Relational Topic Modeling via Graph Poisson Gamma Belief Network

**Chaojie Wang**,*    **Hao Zhang**,*    **Bo Chen**,†    **Dongsheng Wang**,    **Zhengjue Wang**
National Laboratory of Radar Signal Processing
Xidian University, Xi'an, Shaanxi 710071, China
`xd_silly@163.com, zhanghao_xidian@163.com, bchen@mail.xidian.edu.cn`
`wds_dana@163.com, zhengjuewang@163.com`

**Mingyuan Zhou**
McCombs School of Business
The University of Texas at Austin
Austin, TX 78712, USA
`mingyuan.zhou@mccombs.utexas.edu`

## Abstract

To analyze a collection of interconnected documents, relational topic models (RTMs) have been developed to describe both the link structure and document content, exploring their underlying relationships via a single-layer latent representation with limited expressive capability. To better utilize the document network, we first propose graph Poisson factor analysis (GPFA) that constructs a probabilistic model for interconnected documents and also provides closed-form Gibbs sampling update equations, moving beyond sophisticated approximate assumptions of existing RTMs. Extending GPFA, we develop a novel hierarchical RTM named graph Poisson gamma belief network (GPGBN), and further introduce two different Weibull distribution based variational graph auto-encoders for efficient model inference and effective network information aggregation. Experimental results demonstrate that our models extract high-quality hierarchical latent document representations, leading to improved performance over baselines on various graph analytic tasks.

## 1 Introduction

A wide variety of network data, such as citation networks [1], chemical molecular structures [2], and social networks [3], can be represented as a graph composed of a set of objects (nodes), each of which is characterized by a set of node features (attributes), and their relationships (edges). The node features and edges are usually characterized as count, binary, or positive variables. In many graph analytic applications, such as link prediction, modeling uncertainty [4] in the latent space rather than only providing deterministic node embeddings, is of crucial importance [5–7] and can be realized via probabilistic generative models (PGMs).

Various network decomposition methods have been proposed to discover the underlying relationships of the nodes from the link structure [8–11]. These methods, however, often ignore the information provided by the node features. Inspired by the efficiency of latent Dirichlet allocation (LDA) [12] in exploring the hidden structure of count-valued data, a series of relational topic models (RTMs) [3, 13–17] were introduced to explore the relationships between the nodes and edges in a latent space.

---

Though achieving appealing performance, these RTMs are limited by their shallow structures that only employ a single-layer latent representation. Although there is a recent trend to develop deep probabilistic topic models [18–20] replacing LDA to provide multi-layer semantic representations, how to construct a hierarchical RTM to capture multi-layer semantic connections and use them together with graph learning remains an open research problem.

From another perspective, variational autoencoder (VAE) [21, 22] was extended for modeling graph-structured data, resulting in a variational graph autoencoder (VGAE) [23] parameterized by graph convolutional networks (GCNs) [24]. Further, focused on task-specific applications, several VGAE-based variations [2, 25–27] are introduced for various graph analytic tasks. To move beyond the naive choice of a Gaussian prior in combination with the inner product decoder in VGAE, some methods [4, 28] attempt to learn non-Gaussian latent representations and have achieved promising performance. However, these VGAEs mentioned only capture single-layer semantic representations and ignore the reconstruction information of node features, which may hurt the performance on node clustering or classification.

Inspired by both the advantages of existing RTMs and VGAEs and to move beyond their constraints, we first construct an interpretable hierarchical (deep) RTM, which is further developed as two different non-Gaussian VGAEs with multiple stochastic layers. The main contributions of this paper are:

- A novel RTM named graph Poisson factor analysis (GPFA), equipped with analytic conditional posteriors for efficient Gibbs sampling, is proposed to account for both node features and link structure by sharing their latent representations (topic proportions).
- To explore hierarchical latent representations of the nodes and reveal their relationships at different semantic levels, GPFA is extended to a deep generative model, referred to as graph Poisson gamma belief network (GPGBN). To the best of our knowledge, GPGBN is the first unsupervised deep RTM for analyzing network data.
- To move beyond Gaussian-based VGAEs, which often fail to well approximate sparse, nonnegative, and skewed document latent representations, and generalize GPGBN to different tasks, we combine GPGBN (decoder) with two Weibull distribution-based graph variational inference networks (encoder), resulting in two different Weibull graph autoencoders.
- Besides achieving state-of-the-art or comparable performance on various graph analytic tasks, our models provide a potential solution to explore multi-layer interpretable network relationships.

## 2 Related work

Probabilistic representation learning for network data has drawn considerable attentions. The related work can be roughly divided into two categories: one constructs a probabilistic relational topic model while the other leverages a graph autoencoder.

**Relational topic models:** Derived from the traditional topic models [12, 29], RTMs [14, 15] are introduced to jointly consider the generations of document contents and their relationships. Specifically, each document exhibits a latent mixture of topics, while the connections between documents are modeled as binary variables dependent on the topic assignments of the word tokens. Further, borrowing similar ideas from RTMs, more sophisticated probabilistic generative models (PGMs) [30–32] are developed for jointly modeling networks and text with topic models. These PGMs can also be easily applied to other fields, in particular, recommender systems [33, 34]. Due to the non-conjugacy between the prior and the link likelihood, traditional RTMs employ the variational inference (VI) with the mean-field assumption, which is often too restrictive in practice. To alleviate this issue, more sophisticated RTMs [35, 36] are developed for efficient collapsed Gibbs sampling, taking advantages of data augmentation techniques. However, these RTMs under-exploit hierarchical semantics, because $i$) they only employ single-layer document representations and $ii$) the connections are measured based on their shallow representations.

**Graph autoencoders:** Recently, graph neural networks (GNNs) [37] have proven their efficacy in exploring the relational structure among objects, inspiring research involving transforming the document relational network as a graph and using GNNs to learn document representations. Notably, the edges in a graph can be regarded a special attention mechanism, which has been applied for many popular fields and achieved great success [38–41]. Further, VGAE [23] extends VAE [21] for graph-structured data, which encodes both the node features and adjacency matrix using a Gaussian inference model parameterized by GCNs [24], and then decode the latent representation to generate

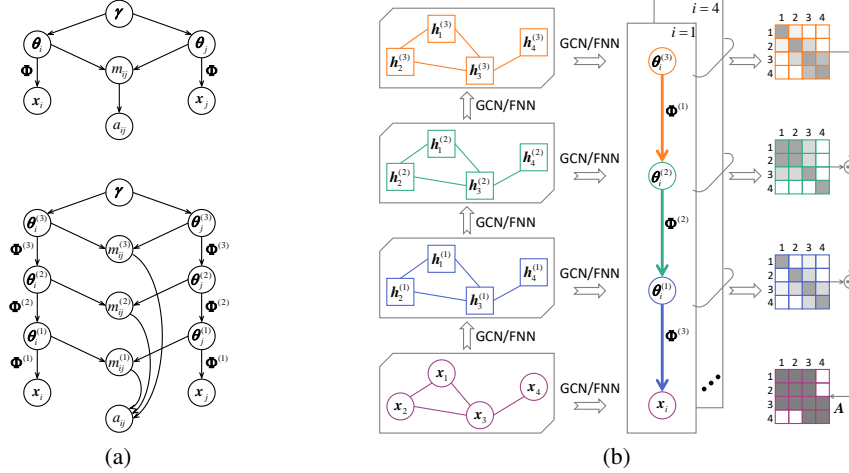

Figure 1: Illustration of (a) the generative (decoder) graphical model and (b) the inference (encoder) network, where the upper one in (a) describes GPFA and the bottom one describes GPGBN.

the adjacency matrix. To enrich the flexibility of posterior distributions, the semi-implicit VI (SIVI) [4, 42] and von Mises-Fisher distribution [28] are introduced to help better approximate complex posteriors. Though achieving state-of-the-art performance, these methods under-interpret the latent structure of network data, *e.g.*, the hierarchical semantics of document contents and the hierarchical relationships between nodes. Moreover, they ignore the generation of node features, resulting in relative poor performance in node-level tasks.

## 3   Deep relational topic models

To investigate the latent structure of network data more intuitively, in the following discussion, we focus on the widely used document relation networks (DRNs), due to their interpretable semantics. Nevertheless, our models can be generalized to other networks with nodes and edges expressed by count, binary, or positive values, as discussed in Appendix A. Below we first discuss how to represent a DRN as a graph and then introduce GPFA, which will be further extended to GPGBN to explore the hierarchical semantic topics and relationships of document networks.

### 3.1   Representing a DRN as a graph

Usually, a DRN containing $N$ documents can be denoted as an undirected graph $\mathcal{G} = \{\mathcal{V}, \mathcal{E}\}$, where $\mathcal{V}$ denotes the set of document nodes and $\mathcal{E}$ denotes the set of edges. Next we describe how to model the node features (document representations) and edges (relationships between documents).

**Node features:**   Due to the efficiency of global semantics, such as word co-occurrence patterns, most DRNs consider the bag-of-words (BoW) vectors as document representations. Formally, the BoW representation of the $i$-th document is a high-dimensional sparse count vector $\boldsymbol{x}_i \in \mathbb{Z}^{K_0}$, where $\mathbb{Z} = \{0, 1, ...\}$ and $K_0$ is the vocabulary size. Naturally, the node features of $\mathcal{G}$ can be represented as a matrix $\mathbf{X} = [\boldsymbol{x}_1, \cdots, \boldsymbol{x}_N] \in \mathbb{Z}^{K_0 \times N}$.

**Adjacency matrix:**   The weighted links in the DRN can be encapsulated as an $N \times N$ adjacency matrix $\mathbf{A} = \{a_{ij}\}_{i=1,j=1}^{N,N}$, where $a_{ij}$ represents the relationship between document $i$ and $j$, taking binary, count, or real positive values [5, 14]. Considering most existing DRNs use a binary link to model whether two documents are associated, we focus on this scenario in the following discussion.

### 3.2   Graph Poisson factor analysis

To construct a probabilistic model for graph $\mathcal{G}$, we choose to model the node-feature likelihood $p(\mathbf{X}|\cdot)$ and edge likelihood $p(\mathbf{A}|\cdot)$ jointly. Specifically, we present an overview of GPFA, as sketched in

Fig. 1(a), to describe the generations of both node features and edges, expressed as

$$\boldsymbol{x}_i \sim \mathrm{Pois}(\boldsymbol{\Phi}\boldsymbol{\theta}_i), \qquad \boldsymbol{\phi}_k \sim \mathrm{Dir}(\eta), \qquad \boldsymbol{\theta}_i \sim \mathrm{Gam}(\boldsymbol{\gamma}, 1/c_j),$$

$$a_{ij} = \mathbf{1}_{m_{ij}>0}, \quad m_{ij} \sim \mathrm{Pois}(\sum\nolimits_{k=1}^{K_1} u_k \theta_{ik} \theta_{jk}), \quad u_k \sim \mathrm{Gam}(\alpha_k, 1/\beta_k), \tag{1}$$

where $\boldsymbol{x}_i$ is factorized as the product of the factor loading matrix (topics) $\boldsymbol{\Phi} = [\boldsymbol{\phi}_1, \ldots, \boldsymbol{\phi}_{K_1}] \in \mathbb{R}_+^{K_0 \times K_1}$ and the gamma distributed factor scores (topic proportions) $\boldsymbol{\theta}_i = [\theta_{i1}; \ldots; \theta_{iK_1}] \in \mathbb{R}_+^{K_1}$ under the Poisson likelihood; the Dirichlet prior is applied on $\{\boldsymbol{\phi}_k\}_{k=1}^{K_1}$, for scale identifiability and ease of inference [20]; the binary edge $a_{ij}$ is generated by first drawing a latent count $m_{ij}$ from the Poisson distribution with rate parameter $\sum_{k=1}^{K_1} u_k \theta_{ik} \theta_{jk}$ and then thresholding the count at 1 through the indicator function $\mathbf{1}_{(\cdot)}$; $u_k$ indicates the importance of the $k$-th topic proportion pair in explaining the relations. Note that for count or real positive edges, one may use the Poisson likelihood or gamma-Poisson link [20] to model $a_{ij}$, which makes GPFA or GPGBN more flexible than GVAEs that often only build binary $\mathbf{A}$ via Bernoulli likelihood, as further discussed in Appendix A.

### 3.3 Graph Poisson gamma belief network

To further explore the multilevel semantics of the documents, one straightforward extension of GPFA is to fix the edge generation in (1) but apply a hierarchical prior on the topic proportion $\boldsymbol{\theta}_i$ via the gamma belief network (GBN) [20]. However, the shallow edge generation ignores the relationships implied in multiple semantic levels. Therefore, as shown in Fig. 1(a), we construct a GPGBN with $T$ hidden layer, expressed as:

$$\boldsymbol{x}_j^{(1)} \sim \mathrm{Pois}(\boldsymbol{\Phi}^{(1)}\boldsymbol{\theta}_j^{(1)}), \left\{\boldsymbol{\theta}_j^{(t)} \sim \mathrm{Gam}(\boldsymbol{\Phi}^{(t+1)}\boldsymbol{\theta}_j^{(t+1)}, 1/c_j^{(t+1)})\right\}_{t=1}^{T-1}, \boldsymbol{\theta}_j^{(T)} \sim \mathrm{Gam}(\boldsymbol{\gamma}, 1/c_j^{(T+1)}),$$

$$a_{ij} = 1(\delta_{ij} > 1), \quad \delta_{ij} = \sum\nolimits_{t=1}^{T} m_{ij}^{(t)}, \quad \left\{m_{ij}^{(t)} \sim \mathrm{Pois}(\sum\nolimits_{k=1}^{K_t} u_k^{(t)} \theta_{ik}^{(t)} \theta_{jk}^{(t)})\right\}_{t=1}^{T}, \tag{2}$$

where, $K_t$ denotes the number of topic at layer $t$. Similar with GPFA, we apply a Dirichlet prior on each column of $\boldsymbol{\Phi}^{(t)}$, and a gamma prior on each $u_k^{(t)}$. Intuitively, the node features $\mathbf{X}$ and adjacency matrix $\mathbf{A}$ are tightly coupled by sharing the multilayer topic proportions $\boldsymbol{\Theta} = \{\boldsymbol{\theta}_j^{(t)}\}_{j=1,t=1}^{N,T}$, making it possible to learn the hierarchical node representations and their relationships at multiple semantic levels. By integrating $m_{ij}^{(t)}$ out, the generative process of the adjacency matrix is equivalent to

$$a_{ij} \sim \mathrm{Bernoulli}\left(1 - \exp\left(-\sum_{t=1}^{T}\sum_{k=1}^{K_t} u_k^{(t)} \theta_{ik}^{(t)} \theta_{jk}^{(t)}\right)\right). \tag{3}$$

Further, using the law of total expectation, we have

$$\mathbb{E}[\boldsymbol{x}_j|-] = \left[\prod_{l=1}^{t}\boldsymbol{\Phi}^{(l)}\right]\frac{\boldsymbol{\theta}_j^{(t)}}{\prod_{l=2}^{t} c_j^{(l)}}, \quad \mathbb{E}[a_{ij}|-] = 1 - \exp\left(-\sum\nolimits_{t=1}^{T}\sum\nolimits_{k=1}^{K_t} u_k^{(t)} \theta_{ik}^{(t)} \theta_{jk}^{(t)}\right), \tag{4}$$

which reveals some appealing model properties as described below.

### 3.4 Model properties

**Hierarchical semantic topics:** Eq. (4) implies that the conditional expectation of $\boldsymbol{x}_j$ is a linear combination of the columns in $\prod_{l=1}^{t}\boldsymbol{\Phi}^{(l)}$, with $\boldsymbol{\theta}_j^{(t)}$ viewed as a document-dependent topic proportion. Therefore, $\prod_{l=1}^{t-1}\boldsymbol{\Phi}^{(l)}\boldsymbol{\phi}_k^{(t)}$ can be naturally interpreted as the projection of topic $\boldsymbol{\phi}_k^{(t)}$ to the observation space, providing us a principled way to visualize the topics at multiple semantic levels.

**Multi-layer semantic relationships:** Defining $a_{ij}^{(t)} = \sum_{k=1}^{K_t} u_k^{(t)} \theta_{ik}^{(t)} \theta_{jk}^{(t)}$ and encapsulating $\{a_{ij}^{(t)}\}_{i=1,j=1}^{N,N}$ as $\mathbf{A}^{(t)}$, the matrix $\mathbf{A}^{(t)}$ can be interpreted as an "adjacency matrix" at layer $t$, as shown in Fig. 3. Further, by substituting $\mathbf{A}^{(t)}$ into (4), we have $\mathbb{E}[\mathbf{A}] = 1 - \exp\left(-\sum_{t=1}^{T}\mathbf{A}^{(t)}\right)$, indicating that the relationships in the observation space aggregates the adjacency matrices across all semantic layers. Moreover, the positive scale parameter $u_k^{(t)}$ changes with $k$ and $t$, which implies that the topic proportions (at different layers) make different contributions to each edge generation.

**Effectiveness of BerPo link:** An appealing property of the BerPo link in (3) (as opposed to other link functions for binary observations, such as logistic/probit) is that the inference cost only depends on the number of nonzeros in the observations [11], making it an ideal choice for the problems involving the large-scale graph with sparse edges. Moreover, this excellent property also makes the graph autoencoder to be introduced below convenient to be inferred.

**Analytic posteriors for efficient gibbs sampling:** Compared with existing RTMs [14, 15] that perform approximate inference due to the non-conjugacy of the model, GPGBN provides analytic conditional posteriors for all parameters, which can be inferred efficiently via a Gibbs sampler. Specifically, the sampling update equation for the topic proportion $\boldsymbol{\theta}_j^{(t)}$ is formulated as

$$p(\theta_{jk}^{(t)}|-) \sim \mathrm{Gam}\big(x_{\cdot jk}^{(t)} + \boldsymbol{\phi}_{k:}^{(t+1)}\boldsymbol{\theta}_j^{(t+1)} + \sum_{i\neq j} m_{ijk}^{(t)}, [-\ln(1-p_j^{(t)}) + c_j^{(t+1)} + u_k^{(t)}\sum_{i\neq j}\theta_{ik}^{(t)}]^{-1}\big), \quad (5)$$

where $x_{\cdot jk}^{(t)}$ and $m_{ijk}^{(t)}$ are latent count variables that are independently sampled from the corresponding node feature and relative edges at layer $t$, respectively. See detailed derivations in Appendix B.

# 4 Weibull graph autoencoders

Although the analytic conditional posteriors of GPGBN result in an efficient Gibbs sampler that can be further accelerated with GPU [43], GPGBN is limited by three disadvantages: $i$) characterized by a top-down generative structure, it relies on time-consuming batch sampling when inferring the latent representations; $ii$) as applied in different tasks and datasets, GPGBN has difficulties in balancing the importance of modeling the node features $\mathbf{X}$ and adjacency matrix $\mathbf{A}$; $iii$) restricted by Gibbs sampling, it is not easy to plug in extra side information to extend GPGBN, such as the node labels. To this end, we combine GPGBN (decoder) with two different Weibull distribution-based graph inference networks (encoder), providing two deep Weibull graph autoencoders.

Given the global parameters $\{\boldsymbol{\Phi}^{(t)}, \boldsymbol{u}^{(t)}\}_{t=1}^T$ (with $\boldsymbol{u}^{(t)} = [u_1^{(t)}, \cdots, u_{K_t}^{(t)}]$) in (14), the task is to infer the local parameters $\boldsymbol{\theta}_j^{(t)}$ and $m_{ij}^{(t)}$. Inspired by $\beta$-VAE [44], we introduce a hyper-parameter $\beta$ into the evidence lowerbound (ELBO) of data log-likelihood $\log p(\mathbf{X}, \mathbf{A})$, which can be expressed as

$$L = \sum_{j=1}^N \mathbb{E}\left[\ln p(\boldsymbol{x}_j|\boldsymbol{\Phi}^{(1)}, \boldsymbol{\theta}_j^{(1)})\right] + \beta\mathbb{E}\left[\ln p(\boldsymbol{A}|\{\boldsymbol{\Theta}^{(t)}\}_{t=1}^T)\right] - \sum_{j=1}^N\sum_{t=1}^T \mathbb{E}\left[\ln \frac{q(\boldsymbol{\theta}_j^{(t)}|-)}{p(\boldsymbol{\theta}_j^{(t)}|\boldsymbol{\Phi}^{(t+1)}, \boldsymbol{\theta}_j^{(t+1)})}\right], \quad (6)$$

where, the expectations are taken with respect to a fully factorized variational distribution as $\prod_{i,j=1}^N\prod_{t=1}^T q(\boldsymbol{\theta}_j^{(t)})q(m_{ij}^{(t)})$, $\beta$ is a trade-off parameter between the two likelihoods, indicating different levels of attentions on nodes and edges. Note that the posterior of $m_{ij}^{(t)}$ follows a Poisson distribution (see details in Appendix B), which is a discrete distribution and hard to optimize. Fortunately, $m_{ij}^{(t)}$ can be integrated out as (3), resulting in that we only need to approximate $q(\boldsymbol{\theta}_j^{(t)})$. This is another reason why we use the BerPo link in GPGBN.

## 4.1 Weibull upward-downward variational encoder

Most existing VGAEs rely on Gaussian latent variables, which often fail to well approximate the posteriors of document latent representations, which are often sparse, nonnegative, and skewed. To circumvent the challenging optimization of the gamma distributed conditional posterior of $\boldsymbol{\theta}_j^{(t)}$ shown in (19), and move beyond deterministic encoder, we adopt a Weibull upward-downward variational encoder (WUDVE) [45] to approximate the gamma distributed conditional posterior as

$$q(\boldsymbol{\theta}_j^{(t)}|-) = \mathrm{Weibull}(\boldsymbol{k}_j^{(t)} + \boldsymbol{\Phi}^{(t+1)}\boldsymbol{\theta}_j^{(t+1)}, \boldsymbol{\lambda}_j^{(t)}), \quad (7)$$

where, the parameters $\boldsymbol{k}_j^{(t)}, \boldsymbol{\lambda}_j^{(t)} \in \mathbb{R}^{K_t}$ are deterministically transformed from the observed node features $\mathbf{X}$ and adjacency matrix $\mathbf{A}$. Below we present two different inference networks to realize those transformations, one based on fully-connected neural networks (FNNs) and the other on GCNs.

**Weibull-based FNN encoder:** To aggregate the information in both node features and edges, a concatenated feature vector $\boldsymbol{d}_j = [\boldsymbol{x}_j; \boldsymbol{a}_j] \in \mathbb{Z}^{K_0+N}$ is constructed for the $j$-th document, where

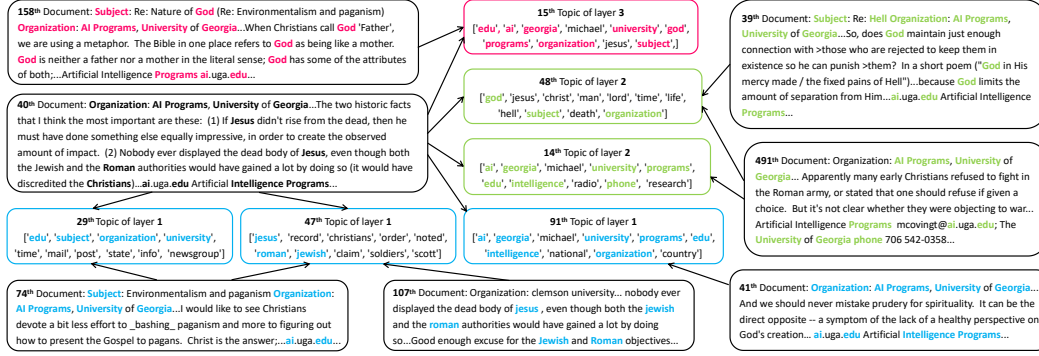

Figure 2: Visualization of a document subnetwork learned by a 3-layer GPGBN on 20news dataset. Taking the 40th document as the source node, other documents, whose connections at layer $t$ (denoted as $u_k^{(t)}\theta_{ik}^{(t)}\theta_{jk}^{(t)}$) are larger than a threshold $\tau$, are displayed in the black boxes, and the key words of connected topics, from shallow to deep, are displayed in the blue, green, and red text boxes, respectively. The common content between the documents and their associated topics are highlighted with the corresponding topic color.

$\boldsymbol{a}_j \in \mathbb{Z}^N$ is the $j$-th column of $\mathbf{A}$, indicating the relationships between the $j$-th document and the other documents. Then, $\boldsymbol{d}_j$ is fed into a FNN to obtain $\{\boldsymbol{k}_j^{(t)}, \boldsymbol{\lambda}_j^{(t)}\}_{j=1,t=1}^{N,T}$ in (7) as

$$\boldsymbol{h}_j^{(0)} = \ln(1 + \boldsymbol{d}_j), \qquad \boldsymbol{h}_j^{(t)} = \ln[1 + \exp(\mathbf{W}_1^{(t)}\boldsymbol{h}_j^{(t-1)} + \boldsymbol{b}_1^{(t)})], \quad t = 1, \cdots, T,$$

$$\boldsymbol{k}_j^{(t)} = \ln[1 + \exp(\mathbf{W}_2^{(t)}\boldsymbol{h}_j^{(t)} + \boldsymbol{b}_2^{(t)})], \quad \boldsymbol{\lambda}_j^{(t)} = \ln[1 + \exp(\mathbf{W}_3^{(t)}\boldsymbol{h}_j^{(t)} + \boldsymbol{b}_3^{(t)})], \quad t = 0, \cdots, T, \quad (8)$$

where $\{\boldsymbol{b}_i^{(t)}\}_{i=1,t=1}^{3,T} \in \mathbb{R}^{K_t}$, $\{\mathbf{W}_i^{(t)}\}_{i=1,t=1}^{3,T} \in \mathbb{R}^{K_t \times K_{t-1}}$, and $\{\boldsymbol{h}_j^{(t)}\}_{j=1,t=1}^{N,T} \in \mathbb{R}^{K_t}$.

**Weibull-based GCN encoder:** Attracted by the excellent ability of GCN [24] in aggregating and propagating graph structure information, we construct a deterministic transformation from $\{\mathbf{X}, \mathbf{A}\}$ to $\{\boldsymbol{k}_j^{(t)}, \boldsymbol{\lambda}_j^{(t)}\}_{j=1,t=1}^{N,T}$ in (7) with GCNs as

$$\mathbf{H}^{(0)} = \ln(\mathbf{X}^T), \qquad \mathbf{H}^{(t)} = \ln[1 + \exp(\tilde{\mathbf{A}}\mathbf{H}^{(t-1)}\mathbf{W}_1^{(t)})], \qquad t = 1, \cdots, T,$$

$$\mathbf{K}^{(t)} = \ln[1 + \exp(\tilde{\mathbf{A}}\mathbf{H}^{(t)}\mathbf{W}_2^{(t)})], \quad \boldsymbol{\Lambda}^{(t)} = \ln[1 + \exp(\tilde{\mathbf{A}}\mathbf{H}^{(t)}\mathbf{W}_3^{(t)})], \quad t = 0, \cdots, T, \quad (9)$$

where, $\mathbf{K}^{(t)} = [\boldsymbol{k}_1^{(t)}; \cdots; \boldsymbol{k}_N^{(t)}] \in \mathbb{R}^{N \times K_t}$, $\boldsymbol{\Lambda}^{(t)} = [\boldsymbol{\lambda}_1^{(t)}; \cdots; \boldsymbol{\lambda}_N^{(t)}] \in \mathbb{R}^{N \times K_t}$, $\{\mathbf{W}_i^{(t)}\}_{i=1,t=1}^{3,T} \in \mathbb{R}^{K_{t-1} \times K_t}$ denotes the GCN filters, $\mathbf{H}^{(t)} \in \mathbb{R}^{N \times K_t}$ the node embedding at layer $t$, $\tilde{\boldsymbol{A}} = \boldsymbol{Q}^{-\frac{1}{2}}\boldsymbol{A}\boldsymbol{Q}^{-\frac{1}{2}}$ the normalized symmetric adjacent matrix which is shared across all layers with degree matrix $\boldsymbol{Q}$.

## 4.2 End-to-end training

Combining the probabilistic decoder GPGBN with the Weibull-based FNN or GCN encoder, we develop two deep probabilistic graph autoencodings based on RTM, as shown in Fig. 1, named as Weibull Graph Autoencoder (WGAE) and Weibull Graph Convolutional Autoencoder (WGCAE), respectively. In what follows, a hybrid Bayesian inference is employed to learn all parameters $\boldsymbol{\Omega} = \{\{\boldsymbol{\Phi}^{(t)}\}_{t=1}^T, \{u_k^{(t)}\}_{k=1,t=1}^{K_t,T}, \mathbf{W}_e\}$ in our models, where $\mathbf{W}_e$ encapsulates all parameters in the encoder. More specifically, we adopt a SGMCMC-based method [46] to update $\boldsymbol{\Phi}^{(t)}$ and a SGD-based method to update $\{\{u_k^{(t)}\}_{k=1,t=1}^{K_t,T}, \mathbf{W}_e\}$, resulting in an end-to-end training. The detailed training algorithm is provided in Appendix C, and the released code[3] is implemented with TensorFlow [47], combined with pyCUDA [48] for parallel Gibbs sampling.

## 5 Experiments

### 5.1 Hierarchical semantic topics and relationships

Distinct from all existing RTMs and GVAEs, GPGBN is able to provide an explicit solution to explore multi-layer interpretable network relationships. To investigate the hierarchical semantic topics and relationships learned by GPGBN, we train a 3-layer GPGBN on 20news dataset with

Table 1: Comparison of node clustering performance.

| Method | Coil | | TREC | | R8 | |
|---|---|---|---|---|---|---|
| | ACC | NMI | ACC | NMI | ACC | NMI |
| NMF [51] | 60.4±0.6 | 72.6±0.5 | 62.6±0.6 | 45.5±0.7 | 54.6±0.6 | 37.4±0.5 |
| LDA [12] | 59.4±0.5 | 71.4±0.5 | 60.9±0.7 | 43.2±0.7 | 53.8±0.6 | 36.9±0.6 |
| PGBN [20] | 61.2±0.3 | 73.4±0.4 | 61.9±0.5 | 44.1±0.4 | 54.7±0.5 | 37.6±0.4 |
| GAE [23] | 65.8±0.4 | 77.0±0.4 | 68.9±0.4 | 52.8±0.3 | 67.2±0.4 | 44.5±0.4 |
| VGAE [23] | 66.3±0.2 | 77.2±0.2 | 69.0±0.3 | 53.0±0.3 | 67.4±0.3 | 44.6±0.3 |
| SIG-VAE [4] | 66.5±0.2 | 77.3±0.2 | 69.2±0.4 | 53.3±0.3 | 67.5±0.3 | 44.8±0.3 |
| RTM [13] | 70.7±0.6 | 82.8±0.5 | 71.5±0.7 | 55.6±0.6 | 70.4±0.7 | 45.9±0.6 |
| GPFA | 73.2±0.5 | 84.6±0.4 | 72.0±0.6 | 56.1±0.6 | 72.4±0.7 | 46.8±0.5 |
| GPGBN | 73.6±0.5 | 85.0±0.5 | 72.3±0.5 | 56.5±0.4 | 73.3±0.5 | 47.5±0.4 |
| GNMF [5] | 78.7±2.5 | 88.2±0.7 | 72.5±1.8 | 56.8±0.9 | 73.8±1.4 | 47.8±1.1 |
| WGAE-layer1 | 77.5±0.4 | 87.7±0.4 | 74.2±0.4 | 58.3±0.3 | 76.4±0.4 | 50.3±0.4 |
| WGAE-layer2 | 79.1±0.2 | 88.1±0.3 | 74.6±0.3 | 58.7±0.2 | 77.3±0.3 | 51.0±0.5 |
| WGAE-layer3 | 81.6±0.2 | 88.3±0.2 | 74.8±0.4 | 59.0±0.2 | 77.6±0.3 | 51.4±0.3 |
| WGCAE-layer1 | 80.5±0.5 | 87.9±0.4 | 74.5±0.4 | 58.6±0.3 | 77.5±0.5 | 51.1±0.4 |
| WGCAE-layer2 | 82.3±0.4 | 89.0±0.4 | 75.0±0.3 | 59.3±0.2 | 78.0±0.4 | 52.0±0.3 |
| WGCAE-layer3 | **83.3**±0.2 | **89.5**±0.2 | **75.3**±0.3 | **59.5**±0.3 | **78.2**±0.3 | **52.3**±0.3 |

efficient Gibbs sampling. As shown in Fig. 2, we select the 40th document as the source node and exhibit other related documents, whose connections to the $i$th ($i = 40$) document satisfy the constraint $u_k^{(t)}\theta_{ik}^{(t)}\theta_{jk}^{(t)} > \tau$, where, $u_k^{(t)}\theta_{ik}^{(t)}\theta_{jk}^{(t)}$ indicates the connection weight between the $i$th and $j$th documents at $k$th topic of layer $t$, $\tau$ is a hyperparameter to adjust the complexity of the subnetwork. Then we display the document contents in the black text boxes and key words of hierarchical topics at multiple semantic levels, from shallow to deep, highlighted in the blue, green, and red text boxes, respectively. It's interesting to notice that the 40th document is related with 41th, 107th, and 74th documents at specific topics of the first hidden layer, like 29th topic on "edu, organization, university" and 47th topic on "jesus, christians, jewish", and has connections with other documents at higher semantic levels. Moreover, there is a clear trend that the topics become more and more general, with the increasing of the network depth, and we will exhibit hierarchical semantic topic trees and more visualized sub-networks learned by GPGBN in Appendix D.

## 5.2  Quantitative graph analysis tasks

The effectiveness and efficiency of WGAE and WGCAE are evaluated on several well-known graph analytic tasks, including link prediction, node clustering, and node classification (in Appendix F).

**Datasets & Model settings:** We consider six widely used benchmarks, including Coil [5], TREC [43], and R8 [49] for node clustering, and Cora, Citeseer and Pubmed [50] for link prediction and node classification. We perform three WGAEs/WGCAEs with different stochastic layers, *i.e.*, $T \in \{1, 2, 3\}$, and set the network structure as $K_1 = K_2 = K_3 = C$, where $C$ is set as the total number of classes for node clustering/classification, and 16 for link prediction following VGAE [23] to make a fair comparision. The summary statistics of these datasets and other implementation details (such as dataset preprocess and hyperparameter settings etc.) are described in Appendix E.

**Node clustering:** We concatenate the multilayer topic proportions $\{\boldsymbol{\theta}_j^{(t)}\}_{t=1}^T$ for each document and employ $K$-means to realize the clustering. Our models are compared with other related node clustering models, mainly concerning three classes: $i$) without edge generation, factorization based methods are considered to model the node features, including NMF [51], LDA [12], and PGBN [20]; $ii$) existing graph autoencoders, GAE, VGAE [23], and SIG-VAE [4], which have no node generation; $iii$) methods that model both node and edge generations, like RTM [13] and GNMF [5].

As shown in Table 1, the accuracy (AC) and normalized mutual information metric (NMI) are used to measure the clustering performance following Cai et al. [5]. Compared with the methods in the first group that only model the node features, the graph autoencoder approaches in the second group aggregate the information in both nodes and edges via the encoder, exhibiting higher scores. However, those graph autoencoders focus obsessively on the generation of the edges but ignore the node generation, leading to worse performance relative to the models in group three, indicating the importance of node generation in clustering tasks. Among these graph-based models in group three, the proposed GPFA outperforms traditional RTM, attributed to the more accurate posterior estimations. With a controllable weight that balances the node and edge generations, WGAEs

Table 2: Comparison of link prediction performance.

| Method | Cora | | Citeseer | | Pubmed | |
|---|---|---|---|---|---|---|
| | AUC | AP | AUC | AP | AUC | AP |
| SC [52] | 84.6±0.01 | 88.5±0.00 | 80.5±0.01 | 85.0±0.01 | 84.2±0.02 | 87.8±0.01 |
| DW [6] | 83.1±0.01 | 85.0±0.00 | 80.5±0.02 | 83.6±0.01 | 84.4±0.00 | 84.1±0.00 |
| GAE [23] | 91.0±0.02 | 92.0±0.03 | 89.5±0.04 | 89.9±0.05 | 96.4±0.00 | 96.5±0.00 |
| VGAE [23] | 91.4±0.01 | 92.6±0.01 | 90.8±0.02 | 92.0±0.02 | 94.4±0.02 | 94.7±0.02 |
| SEAL [7] | 90.1±0.1 | 83.0±0.3 | 83.6±0.2 | 77.6±0.2 | 96.7±0.1 | 90.1±0.1 |
| G2G [53] | 92.1±0.9 | 92.6±0.8 | 95.3±0.7 | 95.6±0.7 | 94.3±0.3 | 93.4±0.5 |
| $S$-VGAE [28] | 94.1±0.1 | 94.1±0.3 | 94.7±0.2 | 95.2±0.2 | 96.0±0.1 | 96.0±0.1 |
| NF-VGAE [4] | 92.4 ±0.6 | 93.0±0.5 | 91.8±0.3 | 93.0±0.8 | 96.6±0.3 | 96.7±0.4 |
| Naive SIG-VAE [4] | 94.0±0.5 | 93.3±0.4 | 94.3±0.8 | 93.6±0.9 | 96.5±0.7 | 96.0±0.5 |
| SIG-VAE (IP) [4] | 94.4±0.1 | 94.4±0.1 | 95.9±0.1 | 95.4±0.1 | 96.7±0.1 | 96.7±0.1 |
| SIG-VAE (K=1, J=1) [4] | 91.8±0.06 | 93.0±0.08 | 91.3±0.04 | 92.4±0.04 | 94.8±0.08 | 95.2±0.06 |
| SIG-VAE (K=15, J=20) [4] | 92.1±0.04 | 93.2±0.06 | 91.6±0.02 | 92.7±0.03 | 95.0±0.08 | 95.4±0.04 |
| SIG-VAE (K=150, J=2000) [4] | **96.0**±0.04 | **95.8**±0.06 | 96.4±0.02 | 96.3±0.02 | **97.0**±0.07 | **97.2**±0.04 |
| WGAE-layer1 | 92.6±0.02 | 93.5±0.03 | 93.6±0.05 | 92.5±0.08 | 94.6±0.04 | 94.8±0.04 |
| WGAE-layer2 | 93.4±0.02 | 94.0±0.02 | 94.1±0.04 | 93.4±0.06 | 95.1±0.02 | 95.3±0.03 |
| WGAE-layer3 | 93.8±0.01 | 94.2±0.02 | 94.3±0.04 | 93.9±0.04 | 95.5±0.03 | 95.8±0.02 |
| WGCAE-layer1 | 93.4±0.04 | 94.1±0.04 | 94.5±0.06 | 94.4±0.07 | 94.9±0.04 | 95.5±0.05 |
| WGCAE-layer2 | 94.5±0.02 | 94.8±0.03 | 95.6±0.03 | 95.8±0.04 | 96.0±0.04 | 96.1±0.04 |
| WGCAE-layer3 | 95.0±0.02 | 95.1±0.02 | **96.5**±0.02 | **96.6**±0.02 | 96.5±0.02 | 96.7±0.02 |

and WGCAEs further improve the clustering performance, where WGCAEs, benefiting from the effectiveness of GCN in graph representation, achieve higher scores than WGAEs under the same network structure settings. Moreover, the performance improvement of PGBN over LDA, GPGBN over GPFA, and multilayer WGAE/WGCAE over single-layer WGAE/WGCAE demonstrate that the richer semantics provided by a deeper probabilistic model can boost the clustering performance.

**Link prediction:** Following VGAE [23], we train the model on an incomplete version of the network data, with $5\%$ and $10\%$ of the citation links used for validation and test, respectively. We realize the link prediction task via link generation and compare our models with some related ones, including spectral clustering (SC) [52], DeepWalk (DW) [6], GAE and VGAE [23], $S$-VGAE [28], SIG-VAE ($K$ and $J$ represents the sampling numbers of SIVI in every iteration), NF-VGAE [4], SEAL [7], and G2G [53].

The comparison results are summarized in Table 2. The binary link prediction task is often regarded as a binary classification task, whose performance is evaluated by the average precision (AP) and area under the ROC curve (AUC), based on 10 random training/testing splits. Compared with traditional graph autoencoder methods (GAE and VGAE), models in the second group provide more flexible posterior distributions than a Gaussian one, achieving better performance. Distinct from these sophisticated methods, such as the SEAL that constructs deterministic representations and the G2G that assumes a Gaussian latent space, the proposed WGAEs and WGCAEs provide more sparse latent document representations and introduce node feature likelihood into the loss function, both of which effectively alleviate overfitting and oversmoothing. Moreover, our models are proficient in exploring hierarchical uncertainties and capturing the relationships at multiple semantic layers, achieving state-of-the-art or comparable performance. Compared to SIG-VAE, which requires setting large $K$ and $J$ that consumes a large memory footprint to achieve the state-of-the-art performance, our models that require much less memory to run are more convenient to be implemented on personal platforms. More detailed comparisons between our models and SIG-VAE are provided in Appendix G.

**Visualization:** As discussed in Section 3.4, the proposed models are able to discover multiple semantic relationships at different hidden layers. Specifically, after training, the observed adjacency matrix $\mathbf{A}$ can be "divided" into multiple adjacency matrices at different semantic layers $\{\mathbf{A}^{(t)}\}_{t=1}^T$. After training a 3-layer WGCAE on Cora and Citeseer, respectively, we randomly select 25 nodes to exhibit their adjacency matrix $\mathbf{A}$ (see the first column of Fig. 3), and the corresponding $\{\mathbf{A}^{(t)}\}_{t=1}^T$ (see columns 2-4 in Fig. 3). For better demonstration, we highlight a part of the link structures with a red bounding box. Clearly, the observed relationships are decomposed across different semantic layers. For example, the red box in Fig. 3(a) highlights four node-node connections, where the first one (from top to bottom) is mainly captured by the third semantic layer, but almost ignored by the second semantic layer. Interestingly, the connection at the bottom right corner of red box in Fig. 3(e) disappears at the first hidden layer but reoccurs at the second and third layers, which illustrates the effectiveness of exploring the relationships in multiple semantic layers rather than a single one.

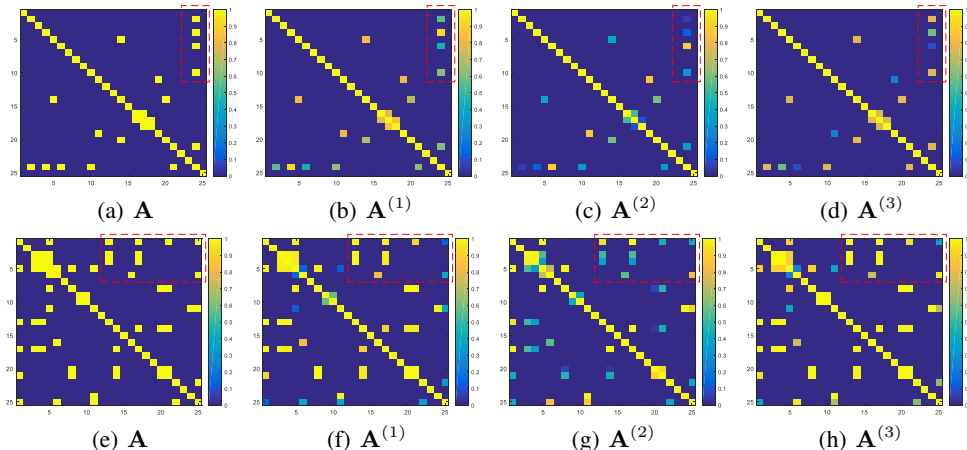

|       |       |       |       |
|-------|-------|-------|-------|
| (a) $\mathbf{A}$ | (b) $\mathbf{A}^{(1)}$ | (c) $\mathbf{A}^{(2)}$ | (d) $\mathbf{A}^{(3)}$ |
| (e) $\mathbf{A}$ | (f) $\mathbf{A}^{(1)}$ | (g) $\mathbf{A}^{(2)}$ | (h) $\mathbf{A}^{(3)}$ |

Figure 3: Visualization of part (randomly selected 25 nodes) of the hierarchical relationships learned by 3-layer WGCAEs on Cora (the first row) and Citeseer (the second row). The first column represents the observed adjacency matrix $\mathbf{A}$ and the second to fourth columns represent the learned adjacency matrices $\mathbf{A}^{(t)}$ from the layer 1 to 3, respectively. After normalization, a brighter point of each $\mathbf{A}^{(t)}$ indicates a stronger node relationship, and the red zone is highlighted for better demonstrations.

## 5.3 Balance between the node and edge generations

As discussed in Section 4, despite the analytic conditional posteriors of GPGBN leading to efficient Gibbs sampling, it is still difficult to control the trade-off between the node and edge likelihoods for different tasks, which limits the modeling capabilities of the proposed GPGBN. On the contrary, moving beyond treating node features $\mathbf{X}$ and adjacency matrix $\mathbf{A}$ equally, WGAE and WGCAE are more flexible via introducing Weibull inference networks, resulting in a controllable weight to balance the focuses on nodes and edges. Notably, we emphasize that the Weibull inference network of either WGAE or WGCAE only approximates the posteriors of latent document representations and can't directly improve the model performance. To evaluate the effectiveness of the weight $\beta$, we perform experiments with different values of $\beta$, and observe the model performance on node clustering and link prediction tasks using the Coil dataset. In Fig. 4, we can see that the best performance of node clustering and link prediction are achieved around $\beta = 0.1$ and $\beta = 100$, respectively. Recapping (6), a larger $\beta$ contributes to more attentions on edge reconstructions rather than node representations. This phenomenon verifies the different contributions of node and edge generations in different tasks. Besides, it also potentially explains the reason why VGAE and SIG-VAE achieve good performance on link prediction but don't work well on node clustering, since they only consider the edge generation.

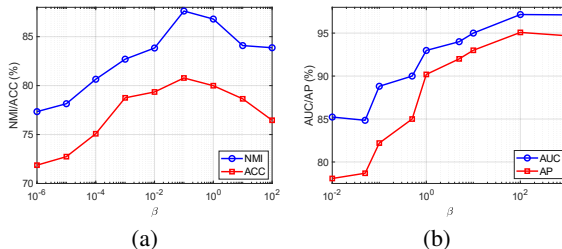

Figure 4: The effect of $\beta$ on (a) node clustering task and (b) link prediction task.

## 6 Conclusion

We propose graph Poisson gamma factor analysis (GPFA) as a new relational topic model, which models both node features (attributes) and link structure (edges) jointly by sharing the topic proportions (latent representations). Then we extend the GPFA to a deep graph Poisson gamma belief network (GPGBN), which is able to explore hierarchical relationships of interconnected documents. Besides performing model inference using analytic conditional posteriors, we further interpret GPGBN as a decoder, and construct two different Weibull distribution based graph encoders, leading to two deep graph autoencoders. Through both qualitative and quantitative experiments, our models are shown to achieve promising results on various graph analytic tasks.

## Broader Impact

The proposed GPGBN can be used to analyze network data, such as citation networks and social networks. Distinct from traditional network analysis, our model can provide intuitive visualization for hierarchical semantic topics and relationships, which potentially explain the underlying reasons for connections between the nodes (representing documents, persons, or other entities) of the network.

The developed WGAE and WGCAE are more flexible for downstream network analysis tasks, like link prediction (predict if there is a connection between the suspects), node classification (determine which community the person belongs to) and so on. Meanwhile, benefiting from incorporating the GPGBN as a decoder, both WGAE and WGCAE can provide interpretable visualization and help the user to explain the basis for the network decision. Of course, these characteristics can also be exploited by ill-intentioned users, so the risk of the proposed models being used in malicious ways cannot be ignored.

We advocate that researchers in this field pay more attention to the study of interpretable graph models, rather than only focusing on the numerical performance. The interpretable model enables the users to understand what the model really learns, which helps to evaluate the trust of the model decision and further explore additional applications.

## Acknowledgements

B. Chen acknowledges the support of the Oversea Talent by Chinese Central Government, the 111 Project (No. B18039), NSFC (61771361) and Shaanxi Innovation Team Project. M. Zhou acknowledges the support of Grant IIS-1812699 from the U.S. National Science Foundation.

## Footnotes

[3]https://github.com/BoChenGroup/GPGBN

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
