[Supplementary Material]

## A Link functions for different types of observations

In this paper, for decoder, we first develop GPFA, which is a shallow topic relational model (RTM) equipped with analytic conditional posteriors, and further extend it to GPGBN, a hierarchical (deep) generalization. For brevity, in the main manuscript, we only give a detailed discussion on the binary type of adjacency matrix $\mathbf{A}$ and a short introduction on how to build other types of $\mathbf{A}$. Thus, in this section, we will provide more detailed discussion on the continuous nonnegative and count-valued types of $\mathbf{A}$, which are also usually used in practice besides the binary type. Considering the likelihood of $\mathbf{A}$ is similar in GPFA and GPGBN, in the following, we take the observed adjacency matrix $\mathbf{A}$ in GPFA as an example to discuss. For completeness, we first discuss the binary $\mathbf{A}$ and then extend it to the continuous nonnegative and count-valued types.

**Binary adjacency matrix:** If the edge $a_{ij}$ is binary to model whether two documents are associated, we can use a Bernoulli-Poisson (BerPo) link [11] expressed as

$$a_{ij} = 1(m_{ij} > 0), m_{ij} \sim \text{Pois}(\sum\nolimits_{k=1}^{K_1} u_k \theta_{ik} \theta_{jk}) \tag{10}$$

where $a_{ij} = 1$ if $m_{ij} \geq 1$ and $a_{ij} = 0$ if $m_{ij} = 0$. After $m_{ij}$ is marginalized out, we can obtain a Bernoulli random variable as $a_{ij} \sim \text{Bern}(1 - \exp(-\sum_{k=1}^{K_1} u_k \theta_{ik} \theta_{jk}))$. The conditional posterior of the latent count $m_{ij}$ can be expressed as

$$(m_{ij} \mid a_{ij}, \boldsymbol{u}, \boldsymbol{\theta_i}, \boldsymbol{\theta_i}) \sim a_{ij} \cdot \text{Pois}_+(\sum\nolimits_{k=1}^{K_1} u_k \theta_{ik} \theta_{jk}), \tag{11}$$

which can be simulated with a rejection sampler as described in Zhou et al. [11].

**Continuous nonnegative adjacency matrix:** If the edge $a_{ij}$ is a continuous nonnegative value, like the cosine similarity between two documents, we can use gamma-Poisson link [20] expressed as

$$a_{ij} \sim \text{Gam}(m_{ij}, 1/c), m_{ij} \sim \text{Pois}(\sum\nolimits_{k=1}^{K_1} u_k \theta_{ik} \theta_{jk}) \tag{12}$$

whose distribution has a point mass at $a_{ij} = 0$ and is continue for $a_{ij} > 0$. Further the latent count $m_{ij} = 0$ if and only if $a_{ij} = 0$, and $m_{ij}$ is a positive integer drawn from a truncated Bessel distribution if $a_{ij} > 0$.

**Count-valued adjacency matrix:** If the edge $a_{ij}$ is a count value, such as indicating the frequency of citation, we can directly factor the discrete value $a_{ij}$ under poisson likelihood as

$$a_{ij} \sim \text{Pois}(\sum\nolimits_{k=1}^{K_1} u_k \theta_{ik} \theta_{jk}) \tag{13}$$

To a conclusion, our models can be generalized to different types of networks with nodes and edges expressed by count, binary, or positive values via making full use of different link functions.

## B Derivation for GPGBN

Here we describe the detailed derivation for graph Poisson gamma belief network (GPGBN) with $T$ hidden layers, expressed as

$$\boldsymbol{x}_j^{(1)} \sim \text{Pois}(\boldsymbol{\Phi}^{(1)} \boldsymbol{\theta}_j^{(1)}), \left\{ \boldsymbol{\theta}_j^{(t)} \sim \text{Gam}(\boldsymbol{\Phi}^{(t+1)} \boldsymbol{\theta}_j^{(t+1)}, 1/c_j^{(t+1)}) \right\}_{t=1}^{T-1}, \boldsymbol{\theta}_j^{(T)} \sim \text{Gam}(\boldsymbol{\gamma}, 1/c_j^{(T+1)}),$$

$$a_{ij} = 1(\delta_{ij} > 1), \quad \delta_{ij} = \sum\nolimits_{t=1}^{T} m_{ij}^{(t)}, \quad \left\{ m_{ij}^{(t)} \sim \text{Pois}(\sum\nolimits_{k=1}^{K_t} u_k^{(t)} \theta_{ik}^{(t)} \theta_{jk}^{(t)}) \right\}_{t=1}^{T}, \tag{14}$$

### B.1 Property

During the inference procedure, we adopt the variable augmentation and marginalization techniques of PGBN [20], exploiting the following properties of the Poisson, gamma, and related distributions:

**Property 1 (P1):** If $x. = \sum\limits_{n=1}^{N} x_n$, where $x_n \sim \text{Poisson}(\theta_n)$ are independent Poisson-distributed random variables, then we have $(x_1, \ldots, x_N) \sim \text{Multinomial}\left(x, \frac{\theta_1}{\sum_{n=1}^{N} \theta_n}, \ldots, \frac{\theta_N}{\sum_{n=1}^{N} \theta_n}\right)$ and $x. \sim \text{Poisson}\left(\sum_{n=1}^{N} \theta_n\right)$.

**Property 2 (P2):** If $x \sim \text{Poisson}(c\theta)$, where $c$ is a constant and $\theta \sim \text{Gam}(a, 1/b)$, then we can marginalize out $\theta$ and obtain $x \sim \text{NB}\left(a, \frac{c}{b+c}\right)$, satisfying a negative binomial (NB) distribution.

**Property 3 (P3):** If $x \sim \text{NB}(a, p)$ and $l \sim \text{CRT}(x, a)$ is a Chinese restaurant table-distributed random variable, then $x$ and $l$ are equivalently jointly distributed as $x \sim \text{SumLog}(l, p)$ and $l \sim \text{Poisson}(-\ln(1-p)a)$ [54].

## B.2 Gibbs sampling

Gibbs sampling is a typical training algorithm for Bayesian models [12, 54, 20], and is applicable when there exist local conjugacies for latent variables, whose conditional distributions will then become tractable and simple to sample from, even though the posterior of the joint distribution of these variables is often intractable. Consistent to these aforementioned Bayesian models [12, 54, 20, 55], the proposed GPGBN provides analytic posteriors for all latent variables and the Gibbs sampling can be naturally applied, which have been proven efficient and can be further accelerated with GPU [43].

Making full use of these properties, we can obtain the following Poisson likelihoods for each hidden layer of GPGBN, formulated as:

$$x_{k_{t-1}j}^{(t)} \sim \text{Pois}(-\ln(1-p_j^{(t)}) \sum_{k=1}^{K_t} \phi_{k_{t-1}k}^{(t)} \theta_{jk}^{(t)}), \tag{15}$$

$$m_{ij}^{(t)} \sim \text{Pois}(\sum_{k=1}^{K_t} u_k^{(t)} \theta_{ik}^{(t)} \theta_{jk}^{(t)}), \tag{16}$$

Zhou2015Negative where the augmented node feature vector $\boldsymbol{x}_j^{(t)} \in \mathbb{R}_+^{K_{t-1}}$ and adjacency matrix $\boldsymbol{M}^{(t)} \in \mathbb{Z}^{N \times N}$ can be regarded as the specific observations at layer $t$. Thus, for each hidden layer of GPGBN, the analytic conditional posteriors for model parameters can be formulated as follows:

**Sampling loading factor matirx $\boldsymbol{\Phi}^{(t)}$:** Utilizing the simplex constraint on each column of $\boldsymbol{\Phi}^{(t)} \in \mathbb{R}_+^{K_{t-1} \times K_t}$, we can marginalize out $\boldsymbol{\theta}_j$ in (15) and have

$$(x_{1jk_t}^{(t)}, ..., x_{K_{t-1}jk_t}^{(t)}|x_{\cdot jk_t}^{(t)}) \sim \text{Multi}(x_{\cdot jk_t}^{(t)} \mid \boldsymbol{\phi}_{k_t}^{(t)}). \tag{17}$$

Therefore, via the Dirichlet-multinomial conjugacy, the posterior of $\boldsymbol{\phi}_{k_t}^{(t)}$ can be formulated as

$$(\boldsymbol{\phi}_{k_t}^{(t)} \mid -) \sim \text{Dir}(x_{1 \cdot k_t}^{(t)} + \eta^{(t)}, ..., x_{K_{t-1} \cdot k_t}^{(t)} + \eta^{(t)}). \tag{18}$$

**Sampling topic proportions $\boldsymbol{\theta}_j^{(t)}$:** Benefit from jointly modeling node features and link structure under poisson likelihood, we have

$$(\theta_{jk_t}^{(t)} \mid -) \sim \text{Gam}\big(x_{\cdot jk_t}^{(t)} + \boldsymbol{\phi}_{k_t:}^{(t+1)} \boldsymbol{\theta}_j^{(t+1)} + \sum_{i \neq j} m_{ijk_t}^{(t)}, [-\ln(1-p_j^{(t)}) + c_j^{(t+1)} + u_{k_t}^{(t)} \sum_{i \neq j} \theta_{ik_t}^{(t)}]^{-1}\big). \tag{19}$$

where $x_{\cdot jk_t}^{(t)}$ and $m_{ijk_t}^{(t)}$ are latent count variables that are independently sampled from the corresponding node feature and relative edges at layer $t$, respectively.

**Sampling scale parameter $c_j^{(t)}$ and $p_j^{(t)}$:** To construct a hierarchical generative model, we introduce $c_j^{(t)} \sim \text{Gam}(e_0, 1/f_0)$ and the corresponding posterior of $c_j^{(t)}$ can be formulated as

$$(c_j^{(t)} \mid -) \sim \text{Gam}(\theta_{j\cdot}^{(t)} + e_0, 1/[f_0 + \theta_{j\cdot}^{(t-1)}]). \tag{20}$$

Referring the Lemma **1** of 20, $\{p_j^{(t)}\}_{t \geq 2}$ can be calculated with

$$p_j^{(t+1)} := -\ln(1-p_j^{(t)})/[c_j^{(t+1)} - \ln(1-p_j^{(t)})], \tag{21}$$

specifically defining $p_j^{(1)} := 1 - e^{-1}$.

**Sampling $\boldsymbol{u}^{(t)}$:** With the prior $u_{k_t}^{(t)} \sim \text{Gam}(\alpha_0, 1/\beta_0)$, via the gamma-Poisson conjugacy, we have

$$(u_{k_t}^{(t)} \mid -) \sim \text{Gam}(\prod_{i=1}^{N} \prod_{j=1}^{i-1} m_{ijk_t}^{(t)} + \alpha_0, [\beta_0 + \prod_{i=1}^{N} \prod_{j=1}^{i-1} \theta_{ik_t}^{(t)} \theta_{jk_t}^{(t)}]^{-1}). \tag{22}$$

## B.3 Time complexity analysis

Compared to the basic PGBN, the additional time cost for GPGBN is mainly in the procedure of graph augmentation, which can be formulated as

$$(m_{ij}^{(1)}, ..., m_{ij}^{(T)} \mid m_{ij}) \sim \text{Multi}(m_{ij}; \frac{\sum_{k_1=1}^{K_1} u_{k_t}^{(t)} \theta_{ik_1}^{(t)} \theta_{jk_1}^{(t)}}{\sum_{t=1}^{T} \sum_{k_t=1}^{K_t} u_{k_t}^{(t)} \theta_{ik_t}^{(t)} \theta_{jk_t}^{(t)}}, ..., \frac{\sum_{k_T=1}^{K_T} u_{k_t}^{(t)} \theta_{ik_T}^{(t)} \theta_{jk_T}^{(t)}}{\sum_{t=1}^{T} \sum_{k_t=1}^{K_t} u_{k_t}^{(t)} \theta_{ik_t}^{(t)} \theta_{jk_t}^{(t)}}),$$
$$(23)$$

$$(m_{ij1}^{(t)}, ..., m_{ijK_t}^{(t)} \mid m_{ij}^{(t)}) \sim \text{Multi}(m_{ij}^{(t)}; \frac{u_1^{(t)} \theta_{i1}^{(t)} \theta_{j1}^{(t)}}{\sum_{k_t=1}^{K_t} u_{k_t}^{(t)} \theta_{ik_t}^{(t)} \theta_{jk_t}^{(t)}}, ..., \frac{u_{K_t}^{(t)} \theta_{iK_t}^{(t)} \theta_{jK_t}^{(t)}}{\sum_{k_t=1}^{K_t} u_{k_t}^{(t)} \theta_{ik_t}^{(t)} \theta_{jk_t}^{(t)}}) \qquad (24)$$

where the former indicates the multi-layer augmentation from $m_{ij}$ and the latter denotes the augmentation at layer $t$. Here we note that there is no need to augment the whole adjacency matrix $\mathbf{A}$, but only the positive elements (edges) $\{a_{ij} > 0\}_{i=1,j=1}^{N,N}$ in $\mathbf{A}$. Thus the computational benefit is significant since the computational complexity is approximately linear to the number of $\{a_{ij} > 0\}_{i=1,j=1}^{N,N}$, denoted as $S_a$, in the observed adjacency matrix $\mathbf{A}$. Compared to directly handling with the whole matrix like traditional RTMs [14], this benefit is especially pertinent in many real-world network data where $S_a$ is significantly smaller than $N^2$. Moreover, the augmentation operations for node features $\mathbf{X}$ or link structure $\mathbf{A}$ can be both processed with parallel Gibbs sampling [43], which can be further accelerated with GPU.

## C Hybrid SGMCMC/VAE inference for WGAE & WGCAE

### C.1 Stochastic gradient MCMC

Although the Gibbs sampler provided in Appendix B can be further accelerated with GPU, there is still a requirement for the GPGBN to preprocess all documents in each iteration and hence has limited scalability. For scalable inference, we consider to update global parameter $\mathbf{\Phi}^{(t)}$ by generalizing TLASGR-MCMC [46, 45], a SGMCMC algorithm that has been applied for the scalable inference for discrete latent variable models (LVMs) [56, 57]. More specifically, TLASGR-MCMC adopts an elegant simplex constraint and increases the sampling efficiency via the use of the Fisher information matrix (FIM), with adaptive step-sizes for the topics of different layers, which can be naturally extended to our model. The efficient TLASGR-MCMC update of $\phi_k^{(t)}$ in GPGBN can be described as

$$\phi_k^{(t)new} = \left\{ \phi_k^{(t)} + \frac{\varepsilon_i^{(t)}}{M_k^{(t)}} [(\rho \widetilde{\boldsymbol{x}}_{:k\cdot}^{(t)} + \boldsymbol{\eta}^{(t)}) - (\rho \widetilde{\boldsymbol{x}}_{\cdot k\cdot}^{(t)} + \boldsymbol{\eta}^{(t)} K_{t-1}) \phi_k^{(t)}] + N\left(0, \frac{2\varepsilon_i^{(t)}}{M_k^{(t)}} \text{diag}(\phi_k^{(t)})\right) \right\}_{\angle}, \quad (25)$$

where $i$ denotes the number of mini-batches processed so far; the symbol $\cdot$ in the subscript denotes summing over the data in a mini-batch; and the definitions of $\rho$, $\varepsilon_i^{(t)}$, $M_k^{(t)}$ and $\{\cdot\}_{\angle}$ are analogous to these in [46] and omitted here for brevity.

### C.2 Hybrid SGMCMC/VAE inference

Combining TLASGR-MCMC and the Weibull upward-downward variational inference network as described in Section 4.1, we can construct a hybrid stochastic-gradient MCMC/autoencoding variational inference for GPGBN. In detail, in each iteration, we adopt TLASGR-MCMC [46] to update $\{\mathbf{\Phi}^{(t)}\}_{t=1}^T$ and the standard Adam [58] to optimize $\{\mathbf{W}_e, \{u_k^{(t)}\}_{k=1,t=1}^{K_t,T}\}$ via maximizing the loss function formulated as

$$L = \sum_{j=1}^{N} \mathbb{E}\left[\ln p(\boldsymbol{x}_j | \mathbf{\Phi}^{(1)}, \boldsymbol{\theta}_j^{(1)})\right] + \beta \mathbb{E}\left[\ln p(\boldsymbol{A} | \{\mathbf{\Theta}^{(t)}\}_{t=1}^T)\right] - \sum_{j=1}^{N} \sum_{t=1}^{T} \mathbb{E}\left[\ln \frac{q(\boldsymbol{\theta}_j^{(t)}|-)}{p(\boldsymbol{\theta}_j^{(t)} | \mathbf{\Phi}^{(t+1)}, \boldsymbol{\theta}_j^{(t+1)})}\right].$$
$$(26)$$

For more efficient computation, the proposed hybrid stochastic-gradient MCMC/autoencoding variational inference algorithm in Algorithm 1, which is implemented in TensorFlow [47], combined with pyCUDA [48].

**Algorithm 1** Hybrid stochastic-gradient MCMC and variational autoencoder inference for WGAE and WGCAE

Set mini-batch size $m \leq N$ and number of layer $T$;

Initialize parameters $\{\{\mathbf{\Phi}^{(t)}\}_{t=1}^{T}, \{u_k^{(t)}\}_{k=1,t=1}^{K_t,T}, \mathbf{W}_e\}$;

**for** $iter = 1, 2, \cdots$ **do**

    For all documents $\mathbf{X} \in \mathbb{Z}^{K_0 \times N}$ and $\mathbf{A} \in \{0,1\}^{N \times N}$ draw random noise $\boldsymbol{\varepsilon} = \{\boldsymbol{\varepsilon}_j^{(t)}\}_{j=1,t=1}^{N,T}$;

    Calculate $\nabla_{\mathbf{W}_e, \boldsymbol{u}} L(\mathbf{W}_e, \boldsymbol{u}; \mathbf{X}, \mathbf{A}, \boldsymbol{\varepsilon})$ according to (25) and update $\{\mathbf{W}_e, \boldsymbol{u} = \{u_k^{(t)}\}_{k=1,t=1}^{K_t,T}\}$;

    Randomly select a mini-batch of $m$ documents to form a subset $\mathbf{X}_s = \{\boldsymbol{x}_j\}_{j=1}^{m}$;

    Sample corresponding topic proportions $\{\boldsymbol{\theta}_j^{(t)}\}_{j=1,t=1}^{m,T}$ to update $\{\mathbf{\Phi}^{(t)}\}_{t=1}^{T}$ according to (6);

**end for**

(a)

(b)

Figure 5: Visualization of document subnetworks learned by a 3-layer GPGBN on 20news dataset. The detailed visualization procedures are analogous to the description in Section 5.1.

# D  Visualizations of hierarchical semantic topics and relationships

## D.1  Visualizations of subnetworks learned by GPGBN

To further investigate the interpretability of our models, we visualize more document subnetworks learned by a 3-layer GPGBN on 20news dataset, as shown in Fig. 5. More specifically, we select the 384th and 46th documents, whose contents focus on different semantic topics, as source nodes and the detailed visualization procedures are analogous to the description in Section 5.1.

Figure 6: Visualization of hierarchical topics learned by 3-layer GPGBNs on 20news dataset.

Table 3: Topic-coherence comprasion of GPGBN and PGBN on 20news dataset.

| Topic layers | hardware | christian | guns | space | graphics |
|---|---|---|---|---|---|
| LDA [12] | 0.530 | 0.561 | 0.491 | 0.538 | 0.564 |
| PFA [54] | 0.494 | 0.560 | 0.483 | 0.520 | 0.555 |
| DPFA [19] | 0.581 | 0.604 | 0.535 | 0.562 | 0.575 |
| PGBN [20] | 0.607 | 0.615 | 0.550 | 0.578 | 0.583 |
| GPGBN | **0.638** | **0.641** | **0.602** | **0.623** | **0.613** |

## D.2 Visualizations of hierarchical semantic topics

Aiming at intuitively visualizing multi-layer semantics captured by GPGBN, we follow PGBN [20] to construct topic trees to visualize the GPGBN learned on 20news dataset. The 20news dataset in our experiments consists documents from 20 different news groups, with a pre-pruned vocabulary of size $K_0 = 2000$. Considering there is no observed adjacency matrix for 20News dataset, we first represent each document as a bag-of-word (BOW) vector to consider global semantics, and then construct the adjacency matrix via measuring the cosine similarity between documents. Setting the hyperparameters same as described in Appendix E, we construct 3-layer GPGBNs with same network structures of $[K_1, K_2, K_3] = [128, 64, 32]$ for different news groups, respectively, and train these GPGBNs with collapsed Gibbs sampling algorithms after 1000 iterations.

As shown in Fig. 6, for each tree, we pick a node at top layer and grow the tree downward by drawing a line from node $k$ at layer $t$, the root or a leaf node of the tree, to node $k'$ at layer $t - 1$ for all $k'$ in the set $\{k' : \mathbf{\Phi}_{k'k}^{(t)} > \tau_t / K_{t-1}\}$ and use $\tau_t$ to adjust the complexity of this tree. For each topic $\phi_k^{(t)}$, we exhibit top-10 word to represent the corresponding semantics and there is a clearly trend that the proposed GPGBN can capture hierarchical semantics with multiple semantic latent representations. More specifically, with the increasing of topic layers, the topic semantics vary from specific to general, due to the fact that higher topics are composed of the lower ones as introduced in Section 3.4.

To make a quantitative comparison between the topics learned GPGBN and other topic models, including LDA [12], PFA [54], DPFA [19] and PGBN [20], we adopt the topic coherence [59], which

measures the semantic coherence in the most significant words (top words) of a topic, as the metric to measure the quality of the learned topics. More specifically, we construct various topic models for each news group of 20news dataset, and then evaluate topic coherence on the top hidden layers of the same size. From the results on different news groups shown in Table. 3, the words among the topics learned by GPGBN are more relevant (or co-occurrence) than those learned by other methods, benefiting from introducing the graph likelihood upon multi-layer document latent representations.

# E   Dataset and model setting

## E.1   Dataset and preprocess

Six widely used datasets are considered in our experiment part, including Coil [5], TREC [43], and R8 [49] for node clustering; Cora, Citeseer and Pubmed [50] for link prediction and node classification. The detailed summary statistics of these benchmark datasets are listed in Table 4.

For node clustering, we follow the preprocess of GNMF [5], constructing a binary adjacency matrix via measuring the consine similarity between node features and then thresholding the matrix with

$$a_{ij} = \begin{cases} 1 & \text{if } \cos(\boldsymbol{x}_i, \boldsymbol{x}_j) \geq \tau \\ 0 & \text{if } \cos(\boldsymbol{x}_i, \boldsymbol{x}_j) < \tau, \end{cases} \tag{27}$$

where $\tau$ is a pre-set hyperparameter and a larger $\tau$ indicates a more sparse adjacency matrix.

For link prediction and node classification, the node features and adjacency matrix are provided by VGAE [23], and we directly follow the preprocess published in their code. To make a fair comparison, we adopt the same train/test/validate split as other methods. More specifically, for link prediction, we train the model on an incomplete version of the network data, with 5%/10% of the citation links used for validation/test and the validation set for node classification is fixed same as the other methods.

Table 4: Statistics of the datasets for various tasks.

| Task | Dataset | Nodes | Edges | Features | Classes |
|---|---|---|---|---|---|
| node clustering | Coil | 1,440 | 4,201 | 1,024 | 20 |
| | TREC | 5952 | 18,013 | 2000 | 8 |
| | R8 | 7674 | 27,513 | 2000 | 8 |
| link prediction or node classification | Cora | 2,708 | 5,429 | 1,433 | 7 |
| | Citeseer | 3,327 | 4,732 | 3,703 | 6 |
| | Pubmed | 19,717 | 44,338 | 500 | 3 |

## E.2   Model setting

**Hyperparameter setting:**   To make an intuitive introduction for hyperparameter settings of our models, we list a tabular overview of parameters names, meanings and corresponding hyperparameter settings in Table 5. Here we note that the following hyperparameter settings are suitable for all our models including GPFA/GPGBN and WGAE/WGCAE.

Table 5: Hyperparameters settings.

| Parameter Priors | Parameter means | Hyperparameters settings |
|---|---|---|
| $\phi_{k_t}^{(t)} \sim \text{Dir}(\eta^{(t)})$ | The $k_t$ topic at layer $t$ | $\eta^{(t)} = 0.01$ |
| $c_j^{(t)} \sim \text{Gam}(e_0, 1/f_0)$ | The scale parameter of $\boldsymbol{\theta}_j$ | $e_0 = 1, f_0 = 1$ |
| $u_{k_t}^{(t)} \sim \text{Gam}(\alpha_0, 1/\beta_0)$ | The importance weight of $\theta_{k_t j}^{(t)}$ | $\alpha_0 = 1, \beta_0 = 1$ |

**Network structure setting:**   We perform three WGAEs/WGCAEs with different stochastic layers, *i.e.*, $T \in \{1, 2, 3\}$, and set the network structure as $K_1 = K_2 = K_3 = C$, where $C$ is set as the total number of classes for node clustering/classification as listed in Table 4, and 16 for link prediction following SIG-VAE [4]. The trade-off parameter $\beta$ in (6) is decided based on the validation data for

Table 6: Comparisons on node classification performances.

| Method | Cora | Citeseer | Pubmed |
|---|---|---|---|
| ManiReg [60] | 59.5 | 60.1 | 70.7 |
| SemiEmb [61] | 59.0 | 59.6 | 71.1 |
| LP [62] | 68.0 | 45.3 | 63.0 |
| DeepWalk [6] | 67.2 | 43.2 | 65.3 |
| ICA [63] | 75.1 | 69.1 | 73.9 |
| Planetoid [64] | 75.7 | 64.7 | 77.2 |
| GCN [24] | 81.5 | 70.3 | 79.0 |
| SIG-VAE [4] | 79.7 | 70.4 | **79.3** |
| GAT-16[4] [65] | 82.3 | 71.9 | 78.7 |
| GAT-64[4] [65] | **83.0** | **72.5** | 79.0 |
| WGCAE | 82.0 | 72.1 | 79.1 |

various graph analysis tasks. For optimization, we adopt the standard Adam [58] with learning rate 0.001 to optimize the whole loss function in (6).

# F   Node classification task

Distinct from the pure probabilistic model GPFA/GPGBN having difficulty to plug in side information, the proposed WGAE/WGCAE are more flexible that can be extended for supervised learning, owing to the VAE-like structure. To make a full investigation for our models, we also perform node classification task with a single-layer WGCAE in this part. Given node labels $\{y_j\}_{j=1}^N$, we incorporate a categorical likelihood $p(y_j \,|\, \boldsymbol{\theta}_j)$ into (6), leading to a supervised model, and the corresponding supervised loss function can be formulated as

$$L_s = \sum_{j=1}^{N} \mathbb{E}\left[\ln p(y_j|\boldsymbol{\theta}_j)\right] + L. \tag{28}$$

To make a fair comparison, we set the hidden units of all methods to 16 following SIG-VAE[4] [4], and the compared results are listed in Table 6. From the node classification results, the proposed WGCAE exhibits strong generalization properties, achieving comparable results on Cora, Citeseer and Pubmed, despite not being trained specifically for node classification task. More specifically, the GCN-based WGCAE outperforms other GCN-based methods and even achieves higher performance than GAT under same network settings. We attribute the advantages of WGCAE to following reasons: $i$) similar to SIG-VAE, the reconstruction for edges and node features effectively alleviate overfitting, which is particularly serious in small datasets like Cora and Citeseer; $ii$) distinct from other methods providing discriminative dense node representations, the WGCAE introduces uncertainty and sparsity into latent representations with Weibull distribution, which is more suitable to approximate sparse and skewed document latent representations.

# G   Discussion about SIG-VAE and WGCAE

We clarify that SIG-VAE [4] and the proposed WGCAE focus on different aspects of improvements, although they are both built on VGAE [23]. SIG-VAE adopts a semi-implicit hierarchical construction, whose first layer is still limited by Gaussian, to support a more complex variational posterior. Moving beyond Gaussian distributions, the proposed WGCAE takes a different approach that uses a Weibull distribution to approximate sparse and skewed gamma distributed conditional posteriors. Moreover, we note that the semi-implicit technique [42] used by SIG-VAE can also be introduced to move WGCAE beyond its Weibull variational distribution, which we leave for future study.

For quantitative comparisons between SIG-VAE and WGCAE, although SIG-VAE outperforms WGCAE in two benchmarks for link prediction, it takes much heavier computation and memory

Table 7: Comparisons on node clustering performances.

| Method | Cora | | Citeseer | |
|---|---|---|---|---|
| | ACC | NMI | ACC | NMI |
| VGAE [23] | 59.2 | 0.43 | 51.5 | 0.20 |
| SIG-VAE [4] | 68.8 | 0.58 | 57.4 | 0.34 |
| WGCAE | **74.6** | **0.62** | **66.5** | **0.41** |

cost. More specifically, for SIG-VAE, if naively implemented, the memory cost is theoretically $(K + J)$ times larger than the most basic VGAE, where $K$ and $J$ represents the sampling numbers of SIVI [32] in each iteration. For example, SIG-VAE takes nearly 0.7G ($K = 1, J = 1$), 4.4G ($K = 5, J = 10$), and 10.6G ($K = 15, J = 20$) RAM cost on Cora with 2708 nodes. Following this trend, a naively implemented SIG-VAE is estimated to take a normally unaffordable memory when setting ($K = 150, J = 2000$). By contrast, a 3-layer WGCAE, which takes only 1.3G RAM, has achieved a comparable link prediction performance and outperformed SIG-VAE on both node clustering and classification tasks, showing the proposed WGCAE is more efficient than SIG-VAE. Further, benefiting from introducing node feature reconstruction, WGCAE significantly outperforms SIG-VAE in node clustering tasks, and we perform additional comparisons between WGCAE and the clustering results provided in the appendix of SIG-VAE [4], as shown in Table 7.

## Footnotes

[4]Note the original GAT uses 64 hidden features (GAT-64), and we also report the results of GAT with 16 hidden features (GAT-16) to make a fair comparison.