[Reviews · NeurIPS 2020]

Review 1

Summary and Contributions: The paper proposes a topic model for networks with text on the nodes (in the vein of the relational topic model). The main innovation is to include a hierarchical structure via a deep probabilistic model by stacking gamma-Poisson factor models, with inference achieved by a sophisticated VAE approach, Weibull graph autoencoders.

Strengths: Although there has been a lot of work on modeling networks with text data via joint topic modeling and network modeling, the use of a deep model in this context is new to this work (as far as I know). This is a useful advance enabling the recovery of complex yet somewhat interpretable models of networks with text on large datasets, with a moderate degree of novelty. The paper builds toward its final approach in several increasingly complex models and inference algorithms, covering a substantial amount of work as a contribution. The proposed model is carefully and insightfully designed to admit tractable and scalable inference. The inference algorithm is principled and leverages recent innovations. This paper is relevant to the NeurIPS community.

Weaknesses: Results were mixed against the strongest baseline, SIG-VAE on the link prediction task, which is arguably the most important evaluation task, performing only slightly better on one out of the three datasets and slightly worse on the other two. The authors claim that the memory requirements are much lower than for this baseline, with further discussion in the appendix. This is fine but the authors should then report hard numbers on memory usage etc to back this claim up. Similarly, the paper states that SIG-VAE "requires an unaffordable memory footprint" but evidently were able to afford it, since they obtained results with that method.

Correctness: As far as I can tell, the claims are correct. I have not verified the finer points of the derivations but it seems fine at a high level. The evaluation is reasonable, with a broad set of baselines and several datasets. Details on how hyper-parameters were selected were in the supplementary - these should probably be moved to the main paper.

Clarity: Yes, the paper is generally well written and well argued. There are a few typos which I will list below.

Relation to Prior Work: While the paper mostly does a good job of contextualizing relative to prior work, there are a couple of issues. There is a substantial literature on jointly modeling networks and text with topic models that this paper only touches on. It would be important to add a section on this, even if it is in the appendix. See for example: Guo, F., Blundell, C., Wallach, H., & Heller, K. (2015, February). The Bayesian echo chamber: Modeling social influence via linguistic accommodation. In Artificial Intelligence and Statistics (pp. 315-323). He, X., Rekatsinas, T., Foulds, J., Getoor, L., & Liu, Y. (2015, June). HawkesTopic: A joint model for network inference and topic modeling from text-based cascades. In International conference on machine learning (pp. 871-880). Zhang, X., & Carin, L. (2012). Joint modeling of a matrix with associated text via latent binary features. In Advances in Neural Information Processing Systems (pp. 1556-1564). Similarly, a few important references to gamma-Poisson factorization models for text or networks should be added: Canny, J. (2004, July). GaP: a factor model for discrete data. In Proceedings of the 27th annual international ACM SIGIR conference on Research and development in information retrieval (pp. 122-129). Gopalan, P., Hofman, J. M., & Blei, D. M. (2013). Scalable recommendation with Poisson factorization. arXiv preprint arXiv:1311.1704. Gopalan, P. K., Charlin, L., & Blei, D. (2014). Content-based recommendations with Poisson factorization. In Advances in Neural Information Processing Systems (pp. 3176-3184).

Reproducibility: Yes

Additional Feedback: Minor suggestions/typos: -In the introduction, prior work is sometimes referred to in the present tense. E.g. "variational autoencoder (VAE) [21, 22] is extended" - "the variational autoencoder ... was extended" would be better. Similarly in several other sentences. Pg 2, "researches" - "research" Pg 6, "the remain documents", "jewis" Pg 7, "balance" - "balances", "whose performance are evaluated", "like the SEAL constructs" - "which constructs", "the G2G assumes" - "the G2G which assumes", "likelyhood", "into loss function" - "into the loss function" Pg 8, "randomly select 25 nodes" - "randomly selected 25 nodes" In Figure 3, it would be more informative to compare these results to a single layer graph Poisson factor model. E.g., can this simpler model adequately capture the adjacency matrix, and is it as interpretable as the proposed model? ----- Post-rebuttal comments: The authors answered my main concern which was the need to provide hard numbers for the memory benefits of the proposed approach. It is important to include these numbers in the revision, as well as the other promised changes. My score will remain unchanged.


Review 2

Summary and Contributions: The paper introduces Graph Poisson-Gamma Belief networks and Weibull Graph Autoencoders to use for relational (graph) data. The proposed three-layer model(s) is better, or on par, with current state-of-the-art performance in multiple numbers of prediction tasks, such as node classification and link prediction.

Strengths: The paper's main benefits are that the new model performs at the same level or outperforms current state-of-the-art problems and is, in many parts, also interpretable. The model also enables the use of the \beta hyperparameter to shift the focus of the model from links (adjacency matrix) to the nodes (textual data). The authors show the problem that models can focus too much on one part, reducing the performance in the other. I also liked the BerPo-link discussion/contribution.

Weaknesses: The main weakness of the paper is the originality of the article. The proposed model is an extension of previously known models to the situation with graph data.

Correctness: The paper seems to be correct in general. I could not spot any direct errors when reading the paper and the supplementary material.

Clarity: The paper is clear, concise, and it is easy to follow. The only limitation is that some parts of the papers are difficult to read due to tiny fonts (e.g., Figure 2 and 4). A similar problem can be found in the appendix.

Relation to Prior Work: Yes, the related work seems to be complete and is discussed in relation to the work.

Reproducibility: Yes

Additional Feedback: The authors propose a Gibbs sampling algorithm that is mentioned to be very efficient. I would expect the parameters to be very correlated, especially in a three-layer model. Could the authors elaborate on this, efficient in what sense? I assume the Gibbs sampler is rather used as a stochastic optimization algorithm than a way to explore the whole posterior? The link activation variable u_k is essentially a variable that will work on the topic level to give strength to individual topics for the links. This variable seems a little bit limiting to me. Have the authors considered a matrix for u instead with u_{k_j,k_i} to better capture interaction effects between different topics? It seems quite straightforward to me and would not increase the number of parameters much? I also miss a discussion on the authors' conclusions about why the Weibull encoder improves so much upon just using the original model with the Gibbs sampler. Since the improvements are quite significant, I think it is essential to discuss why we see the improvement. Where does this additional performance come from? #### AFTER REBUTTAL #### I would like to thank the authors for the rebuttal. I think it is quite clear that the chain is not mixing well (during these 2000 iterations). This is clear in the left plot in Figure 2. Although, I’m not sure this actually matters much as the authors are using the Gibbs sampler - the performance is still good. I think, if the authors are accepted, they should try to clarify what they mean with an efficient Gibbs sampler in the light of Figure 2. I also think the discussion in the rebuttal on why the improvement using the Weibull encoder improves performance. I reread section 5.3 and I do not see that as clear as in the authors’ rebuttal.


Review 3

Summary and Contributions: This paper first proposed Graph Poisson Factor Analysis (GPFA), a relational topic model, to analyze a collection of interconnected documents.This paper also provide a closed form Gibbs sampling approach to approximate the posteriors. Moreover, the paper also proposed GPGBN to further explore the multilevel semantics, with two Weibull distribution based variational graph auto-encoders for efficient model inference and effective network information aggregation.

Strengths: This paper is well-written and easy to read. The authors provide sufficient details for each model, and the experimental part is also convincing.

Weaknesses: It seems the empirical performance of this work compares only favorable with the other baselines. ====post rebuttal============= Thanks for the feedback. The rebuttal addresses my concerns and I will not change my score.

Correctness: Might be correct.

Clarity: Yes

Relation to Prior Work: Yes

Reproducibility: Yes

Additional Feedback:


Review 4

Summary and Contributions: This paper proposes a new document network model, Graph Poisson Gamma Belief Network. Such a probabilistic model can model the document network well and has the ability to capture uncertainty.

Strengths: The paper resolved an important problem in the text mining area. The authors have proposed a new Weibull graph autoencoder which is innovative. The topic is within the scope of NeurlPS.

Weaknesses: The Graph Poisson Factor Analysis and Graph Poisson Gamma Belief Network are straightforward extensions of Mingyuan Zhou's former works on Poisson Gamma Belief Network (NIPS2015). In the experiments, the GCN and other variants of graph nerual networks should be included to show the performance on network modeling. Since the title and main compartive method are both topic model, it would be better to show the evaluation of underlying topics compared to other topic models apart from the link prediction tasks.

Correctness: The calims and method are correct.

Clarity: The paper is well written and easy to follow. The model propeties well explained.

Relation to Prior Work: Yes, it is clear.

Reproducibility: Yes

Additional Feedback:

[Author Response · NeurIPS 2020]

First of all, we thank all the reviewers for their valuable comments and suggestions.

Figure 1: Graph likelihood of GPGBNs as a function of iteration with various network depths.

**To Reviewer 1:** $i$) For SIG-VAE, the memory cost is theoretically $(K + J)$ times
larger than the most basic VGAE, where $K$ and $J$ represents the sampling numbers
of SIVI [32] in each iteration. With the official release, SIG-VAE takes nearly 0.7G
$(K = 1, J = 1)$, 4.4G $(K = 5, J = 10)$, and 10.6G $(K = 15, J = 20)$ RAM
cost on Cora (the smallest dataset with 2708 nodes). Following this trend, SIG-
VAE is estimated to take at least 600G RAM on Cora with their original setting
$(K = 150, J = 2000)$, which is a normally unaffordable memory. By contrast, a 3-
layer WGCAE, which takes only 1.3G RAM, has achieved a comparable link prediction
performance and outperformed SIG-VAE on both node clustering and classification
tasks, showing the proposed WGCAE is more efficient than SIG-VAE. $ii$) Thanks for
pointing out these references, we will carefully investigate them and comprehensively discuss their relations to our
work. We will fix the listed typos in our revision. $iii$) For an intuitive quantitative comparison of network modeling, we
estimate three GPGBNs with different depths $T \in \{1, 2, 3\}$ on 20news dataset and exhibit the likelihood of adjacency
matrix as a function of iterations. As shown in Fig. 1 here, increasing the network depth in general improves the quality
of adjacency matrix fitting, showing the benefit of capturing document relations with a hierarchical structure. Moreover,
we will provide more visualized GPGBNs with different depths in our revision.

**To Reviewers 2&4 (Novelty):** As claimed in our contributions, we propose the first hierarchical relational topic model
(RTM) named GPGBN, and successfully illustrate the connections at different semantic levels. Moreover, our work
provides a novel solution to combine the RTM and graph autoencoders, firstly adopting the GCN to estimate the
posteriors of the latent representations of RTMs (note related theoretical proofs [44] have only recently been proposed).
Moving beyond deterministic projecting, the uncertainty and sparseness provided by Weibull reparameterization
effectively alleviate overfitting of GCN and further improve the performance in a hierarchical fashion.

**To Reviewer 2:** $i$) Gibbs sampling is applicable when there exist local
conjugacies for latent variables, whose conditional distributions will then
become tractable and simple to sample from, even though the posterior of
the joint distribution of these variables is often intractable. Gibbs sampling
uses a Markov chain to sample the latent variables in turn to iteratively
approach the true posteriors. In Fig. 2, we show the trace plot of a random
dimension of $\boldsymbol{u}^{(1)}$ and that of $\boldsymbol{u}^{(2)}$ from a 3-layer GPGBN, suggesting
the Markov chain under the proposed Gibbs sampler converges faster
and mixes well. $ii$) That is a nice idea. We note a full matrix for $U$

Figure 2: Gibbs sampling of variables selected from $\boldsymbol{u}^{(1)}$ and $\boldsymbol{u}^{(2)}$ as a function of iteration.

could provide extra flexibility to model stochastic equivalence/disassortativity (e.g., in protein-protein interaction
network), while a diagonal one is more suitable to model an assortative relational network exhibiting homophily
(e.g., co-author network) but not necessarily stochastic equivalence. Similar conclusion can be found in [34] and we
will add a discussion in Appendix. $iii$) Actually, the Weibull inference network only approximates the posterior of
latent document representation $\boldsymbol{\theta}$ and can't directly improve the performance. However, moving beyond treating the
importance of document content and relations equally like GPGBN, WGCAE is a VAE-like model that can be trained
via optimizing the loss function, where we can introduce a trade-off parameter $\beta$ to control the focus of the model. By
adjusting $\beta$, WGCAE can provide more expressive latent representations for down-stream tasks and we have discussed
this phenomenon in Section 5.3.

**To Reviewer 3:** We clarify that the proposed WGCAE is a basic
VGAE-like model, which has a significant improvement compared to
the original VGAE on various graph analytics tasks. Other relevant
improvement techniques, such as SIVI [32] and GAT [27], can poten-
tially be incorporated into our models to further improve the model
performance; we leave these further extensions for future study.

Table 1: Topic-coherence comparisons on 20news.

| Topic layers | hardware | christian | guns | space | graphics |
|---|---|---|---|---|---|
| LDA [11] | 0.530 | 0.561 | 0.491 | 0.538 | 0.564 |
| PFA [33] | 0.494 | 0.560 | 0.483 | 0.520 | 0.555 |
| AVITM [45] | 0.434 | 0.495 | 0.422 | 0.451 | 0.483 |
| DPFA [19] | 0.581 | 0.604 | 0.535 | 0.562 | 0.575 |
| PGBN [20] | 0.607 | 0.615 | 0.550 | 0.578 | 0.583 |
| GPGBN | **0.638** | **0.641** | **0.602** | **0.623** | **0.613** |

**To Reviewer 4:** $i$) We'd like to emphasize that we have compared with many GCN-based methods in our experiments,
including node clustering (2nd block of Table 1), link prediction (Table 2), and node classification (Table 4 in Appendix).
As far as we know, VGAE could be the most popular GCN-based method for network modeling and other variants like
SIG-VAE, $S$-VGAE, and NF-VGAE have all been included in our comparison. We are also glad to compare with other
VGAEs (if any) for network modeling. $ii$) Thanks for your suggestions, we have included topic-coherence comparison
between hierarchical topics learned by PGBN and GPGBN in Appendix, showing that the words among the topics
learned by GPGBN are more relevant (or co-occurrence) than those learned by PGBN. Moreover, we also add additional
topic-model baselines including LDA, PFA, AVITM [45], and DPFA for topic-coherence comparisons as shown in
Table. 1, indicating the benefit of introducing hierarchical graph regularization. We will put these results in our revision.

[44] Zhao L, Akoglu L. Connecting Graph Convolutional Networks and Graph-Regularized PCA. In ICML, 2020.
[45] Srivastava, A. and Sutton, C. Autoencoding variational inference for topic models. In ICLR, 2017.


[Meta-Review · NeurIPS 2020]

The paper, the reviews, the author response and the ensuing discussion were all taken into consideration. All reviewers considered the work marginally above the acceptance threshold. Novelty was a concern for some but other reviewers appreciated it. Lacking comparisons to GCN and others, evaluation of underlying topics, and consideration of topic modeling prior work were also concerns. However, the paper was generally felt to represent good work, and use of a deep model in this context, design of the model, and convincing experiments were appreciated. Overall the paper seems to be of sufficient quality to be presented at NeurIPS.